# The Emerging Role of Extracellular Vesicles from Mesenchymal Stem Cells and Macrophages in Pulmonary Fibrosis: Insights into miRNA Delivery

**DOI:** 10.3390/ph15101276

**Published:** 2022-10-17

**Authors:** Shuang Li, Jingang Zhang, Guangjiao Feng, Lingmei Jiang, Zhihong Chen, Wenqiang Xin, Xiuru Zhang

**Affiliations:** 1Department of Respiratory Medicine, The Third People’s Hospital of Longgang District Shenzhen, Shenzhen 518112, China; 2Department of Orthopaedic, Jiamusi Central Hospital, 256 Zhongshan Street, Xiangyang District, Jiamusi City 154003, China; 3Department of Respiratory Medicine, Clinical Medical School, Jiamusi University, Jiamusi City 154007, China; 4Department of Neurosurgery, Tianjin Medical University General Hospital, Tianjin 300052, China; 5Department of Pathology, The Third People’s Hospital of Longgang District Shenzhen, 278 Songbai Road, Henggang Street, Shenzhen 518112, China

**Keywords:** pulmonary fibrosis, extracellular vesicles, mesenchymal stem cells, macrophages, microRNAs

## Abstract

Pulmonary fibrosis is a type of chronic, progressive, fibrotic lung disease of unclear cause with few treatment options. Cell therapy is emerging as a promising novel modality for facilitating lung repair. Mesenchymal stem cell (MSC)-based and macrophage-based cell therapies are regarded as promising strategies to promote lung repair, due to incredible regenerative potential and typical immunomodulatory function, respectively. Extracellular vesicles (EVs), including exosomes and microvesicles, are cell-derived lipid-bilayer membrane vesicles that are secreted from virtually every cell and are involved in intercellular communication by delivering expansive biological cargos to recipients. This review provides a deep insight into the recent research progress concerning the effects of MSC and macrophage-associated EVs on the pathogenesis of pulmonary fibrosis. In addition to discussing their respective vital roles, we summarize the importance of cross-talk, as macrophages are vital for MSCs to exert their protective effects through two major patterns, including attenuating macrophage activation and M1 phenotype macrophage polarization. Moreover, miRNAs are selectively enriched into EVs as essential components, and consideration is given to the particular effects of EV-associated miRNAs.

## 1. Introduction

Pulmonary fibrosis is a chronic and progressive parenchymal lung disease characterized by irreversible lung scarring. Pulmonary fibrosis can lead to significant morbidity via aggravating dyspnoea and coughing, as a result causing overall irremediable functional decline [1]. The prevalence of pulmonary fibrosis has increased, and its rates are three to nine cases per 100,000 people annually in Europe and North America, and approximately four cases per 100,000 people annually in East Asia and South America [2]. It predominantly occurs in older adults (>50 years old), with an average life expectancy of 2.5–3.5 years after diagnosis, which is comparable to or even more severe than some cancers [3,4]. Although pirfenidone and nintedanib can promote lung function, neither is curative for the disease [5,6,7]. Cell therapy is emerging as a promising novel modality for facilitating lung repair. Mesenchymal stem cell (MSC) transplantation is attractive therapy since MSCs have the potential to suppress detrimental immune response, promote the survival of injured cells, and enhance tissue repair and regeneration [8,9,10]. MSCs can be obtained from a great number of tissues, such as the umbilical cord, endometrial polyps, menses blood, bone marrow, adipose tissue, etc. [10]. MSCs can interact with cells of the innate and adaptive immune systems, leading to the modulation of several effector functions [10]. Interestingly, macrophages are the key players in organ systems’, including the lungs’, innate and adaptive immune responses to particles and pathogens [11,12]. Macrophages also play vital roles in maintaining homeostasis and pathogen clearance [11]. The classically activated macrophage (M1 phenotype) and alternatively activated macrophage (M2 phenotype) have been extensively investigated in pulmonary fibrosis [12].

MSC-derived extracellular vesicles (EVs) have regulatory properties and transport functional “cargo” through physiological barriers to target cells, for purposes of communication and regulatory activities. Notably, intensive basic experimental and preclinical studies have identified MSC-derived EVs as possible novel therapeutic tools and provided new insights into the treatment of pulmonary fibrosis. As an essential means of intercellular communication, EVs have attracted considerable interest in this context. Compared with direct donor cell transplantation, EV cargos can be protected from degradation by nucleases due to the lipid bilayer membrane that can protect vesicle stability during circulation [13]. Furthermore, EVs show lower immunogenicity and toxicity when compared to other nano-carriers [14]. EV secretion appears to be an evolutionarily conserved procedure, and EV-based cell-to-cell communication occurs throughout all kingdoms of life [15]. EVs are often classified according to their diameter size, as exosomes (30 to 150 nm), microvesicles (50–1000 nm), or apoptotic bodies (500–5000 nm) [16,17,18]. In fact, size, which had been regarded as a decisive factor for distinguishing different vesicles, has been found to be less relevant due to size discrepancies observed by various measuring techniques [19]. EVs are lipid bilayer-enclosed spheres with a critical role in delivering membrane-bound proteins, bioactive metabolites, and RNAs to recipient cells [20]. The use of carrier systems for the delivery of therapeutic payloads to targeted cells has attracted considerable attention, and due to the capacity to shuttle cargos, EVs have gained prominence in the field of nanotherapeutics.

Currently, much insight has been obtained into the non-coding RNAs (ncRNAs) that regulate gene expression at the levels of transcription, post-transcription, and epigenetic processes. Many types of ncRNAs have been categorized as long non-coding RNAs (lncRNAs), microRNAs (miRNAs/miRs), or circular RNAs (circRNA) [21,22]. Interestingly, miRNAs are selectively enriched as essential components in EVs, and miRNAs shuttled in EVs can exert biological functions to regulate specific aspects of the onset and progression of pulmonary fibrosis. This review is intended to summarize the latest literature concerning the effects of MSC and macrophage-associated EVs as therapeutic tools for pulmonary fibrosis, particularly emphasizing the effects of EV-associated miRNAs. A critical discussion is provided of the vital role played by macrophages allowing MSCs to exert their protective effects.

## 2. Ethics Statement

Because this is a review article, an ethics statement is not available. This exemption was agreed by the ethics committee of The Third Affiliated Hospital of Guangzhou Medical University.

## 3. An Overview of the Characteristics of Extracellular Vesicles

EVs cannot replicate, and are steadily secreted from all eukaryotic cells and from various bacteria and archaea species. They have a structure consisting of a shell of a phospholipid bilayer, and appear abundantly in body fluids including blood, urine, breast milk, lacrimal, etc. [23,24]. They are classified into two major subcategories based on biogenesis and size under healthy conditions. Roughly described, microvesicles are 50–1000 nm in diameter and are released by budding directly from the cell plasma membrane [25,26]. Exosomes, the smallest EVs, are single-cell membrane vesicles with a size of approximately 30–150 nm and a density of about 1.13–1.21 g/mL, and have been extensively studied [17,25]. Exosomes originate from the endosomal compartment produced by the curvature of the membrane of the multivesicular bodies (MVBs). Such MVBs cause their initial formation as intraluminal vesicles (ILVs) enclosed in a lipid bilayer [27]. The exosomal membrane contains late endosomal markers (Tsg101, CD63, CD9, CD81, etc.) and origin cell-specific membrane markers [28]. Apoptotic bodies, another kind of EV, are 500–5000 nm in diameter. They are produced by dying cells and are even more abundant than exosomes or MVs under specific conditions [18,29,30]. Exosomes and microvesicles have been suggested as ‘safe containers’ mediating cell-to-cell communication, however, apoptotic bodies are derived from disassembling an apoptotic cell into subcellular fragments. After being secreted into extracellular space, these nanoparticles can naturally target neighboring or distant recipient cells and activate downstream signal pathways in recipient cells by transferring membrane-protected cargos. Among the cytosolic cargos in EVs, ncRNAs have been intensively studied because they play a crucial role at transcriptional and post-transcriptional levels.

## 4. Contributions of EVs Derived from Mesenchymal Stem Cells (MSCs) in the Treatment of Pulmonary Fibrosis

Mesenchymal stem cells (MSCs) transplantation is an attractive therapy. MSCs have the abilities of proliferation and multilineage differentiation, and exhibit immunomodulatory properties [31,32]. Although MSCs can originate from any kind of tissue beyond bone marrow, adipose, and placenta, they have similar core attributes of capacity for cell migration, and behave as a treatment for lung repair [33,34,35]. We conducted a comprehensive literature search of preclinical studies on the effect of EVs derived from MSCs for the treatment of pulmonary fibrosis in lung repair, through various electronic databases including PubMed, Cochrane Library, EMBASE, and Web of Science, from the inception of these databases to 31 August 2022. The following keywords were used in combination with Boolean logic: “EVs”, “extracellular vesicles” or “exosomes” and “stem cell”, together with “pulmonary fibrosis”. Then, the appraised reference list was manually checked to identify other potential qualification trials. The process was iterated until no more publications were obtained. A total of 17 studies were uncovered on this topic, from China, USA, Brazil, Australia, France, and Korea, performed between 2014 and 2022 [36,37,38,39,40,41,42,43,44,45,46,47,48,49,50,51,52]. More information is summarized in Table 1.

### 4.1. EVs from Adipose-Derived MSCs

EVs derived from many MSCs therapies have been extensively studied as available therapeutic strategies in preclinical models of lung diseases; among these MSCs, the therapeutic effects of adipose-derived MSCs and their corresponding EVs have been demonstrated. Interestingly, adipose-derived EVs can be efficiently taken up by alveolar macrophages in vitro and in vivo, serving as a crucial regulator of M2 macrophage polarization by transferring miR-27a-3p to macrophages for repair of acute lung injury [53]. Interestingly, Huang et al. aimed to determine whether EVs from young and aging MSCs had differential effects on lipopolysaccharide-induced acute lung injury in young mice. They revealed that EVs derived from young adipose-derived MSCs were associated with a more significant effect on alleviating tissue injury, due to a lower level of proinflammatory genes; however, EVs of aged cells had no protective response [54]. Silicosis, an occupational disease, appears in patients who inhale silica particles, causing extensive pulmonary fibrosis and ultimately leading to respiratory failure. Bandeira et al. indicated that, adipose MSCs and their EVs, locally delivered at day 30, ameliorated fibrosis and inflammation with reduction in collagen fiber content, size of granuloma, and the number of macrophages inside the granuloma and the alveolar septa [47]. Notably, dose-enhanced EVs yielded better therapeutic outcomes in this model of silicosis [47].

### 4.2. Umbilical Cord MSC-Derived EVs

Of these 17 studies, three evaluated the effect of umbilical cord MSC-derived EVs on the regulation of pulmonary fibrosis [44,45,46]. Shi et al. found that umbilical cord MSC-derived EVs suppressed pulmonary fibrosis and boosted the proliferation of alveolar epithelial cells in fibrosis mice, as a result promoting quality of life, including survival rate, body weight, fibrosis degree, and myofibroblast over-differentiation in lung tissue [46]. These effects of EVs on pulmonary fibrosis were probably achieved by suppressing the transforming growth factor-β (TGF-β) signaling pathway, evidenced by decreased expression levels of TGF-β2 and TGF-βR2. Likewise, Xu et al. showed that EVs from three-dimensional cultured umbilical cord MSC had the potential to repress silica-induced pulmonary fibrosis and improve lung function [45]. Xu et al. further demonstrated that such EVs served as a mediator by transmitting let-7i-5p to decrease fibroblast activation. A decrease in fibroblast activation contributes to repairing pulmonary fibrosis mechanistically via the TGFBR1/Smad3 signaling pathway [44].

### 4.3. Bone Marrow MSC-Derived EVs

Bone marrow is one of the most extensively investigated sources of MSCs, nine of these studies having assessed the therapeutic effect of bone marrow MSC-derived EVs on pulmonary fibrosis [36,37,38,39,40,41,42,43,52]. For example, bleomycin has been used as a primary strategy to induce a pulmonary fibrosis model, and this approach has been well characterized. Bleomycin is always administered by tracheal instillation using an intratracheal aerosolizer. Interestingly, treatment with intravenous human bone marrow MSC-derived EVs sowed a reversal of damage induced by pulmonary fibrosis through reducing collagen deposition. Histopathological observations of lung tissue after injection of EVs from MSCs pretreated in a 3D culture microenvironment found associations with increased collagen deposition, myofibroblast differentiation, and leukocyte infiltration [36]. Moreover, human bone marrow MSC-derived EVs can also be involved in alleviating silica-induced pulmonary fibrosis by reducing the expression of profibrotic factor TGF-β1 and repressing the progression of epithelial-mesenchymal transition [37]. In addition to their effects on silica- and bleomycin-induced pulmonary fibrosis, human bone marrow MSC-derived EVs can inhibit the progression of LPS-induced lung injury and fibrosis by repressing NF-κB and Hedgehog pathways [38]. Of note, mouse bone marrow MSC EVs have a similar effect to human EVs, since mouse EVs appeared in the lungs of systemic sclerosis mice with promotion of anti-inflammatory and antifibrotic markers, while IFNγ pre-activation accelerated the therapeutic effect [43].

### 4.4. Other MSC-Derived EVs

EVs derived from many other MSCs, such as amnion epithelial cells, human embryonic stem cells, human placenta, and menstrual blood-derived endometrial stem cells, have also been involved in alleviating pulmonary fibrosis [48,49,50,51]. Soluble factors derived from human amnion epithelial cells are capable of anti-inflammatory, antifibrotic, and pro-regenerative activity, making them a potential novel therapy for pulmonary fibrosis [48]. Administration of human embryonic stem cell EVs showed a therapeutic effect, attenuating pulmonary fibrosis and improving lung function [49]. Mechanistically, such EVs could exert antifibrotic activity with a reduction of collagen levels during fibrogenesis, at least in vitro, by repressing the TGFβ/Smad pathway. Moreover, human placenta EVs can reduce pulmonary radiation injury by transmitting miR-214-3p, suggesting new avenues to relieve lung injury [51]. Such EVs reduced levels of radiation-induced DNA damage by downregulating ATM/P53/P21 signaling. The downregulation of ATM was regulated by miR-214-3p, which was enriched in EVs. Menstrual blood-derived endometrial stem cell-associated EVs are a new approach to exocytosis in treating fibrotic lung disease, since the discovery that Let-7 from EVs can inhibit pulmonary fibrosis [50]. 

## 5. Contributions of EVs Derived from Macrophages in the Treatment of Pulmonary Fibrosis

Macrophages, the innate immune cells, have antimicrobial phagocytic potential with an essential impact on the pathogenesis of pulmonary fibrosis [55]. Macrophages play a vital role in all stages of lung injury and repair, and especially have the capacity to increase as well as to decrease fibrosis [55]. Lumen-based alveolar macrophages in the airway and parenchymal interstitial macrophages in the lung are two major distinct subtypes. When the lung is insulted and subsequently progresses to pulmonary fibrosis, they are pushed towards a proinflammatory phenotype (M1 macrophages) and eventually polarized to a pro-remodeling phenotype (M2 macrophages) required to ensure restoration of physiological tissue composition [56]. M1 macrophages are proinflammatory cells that regulate extracellular matrix-degrading metalloproteases and proinflammatory cytokines, while M2 macrophages are anti-inflammatory cells that secrete anti-inflammatory cytokines and support tissue healing [57,58]. Nevertheless, extraordinary evidence indicates the importance of macrophage polarization in the modulation of pulmonary fibrosis. M2-polarized macrophages play a crucial role in the progress of pulmonary fibrosis, due to their capability of differentiating into fibrocyte-like cells that produce collagen [57]. For example, in a recent study, bleomycin-induced pulmonary fibrosis was used as a model for pulmonary fibrosis and Schisandra, a commonly used traditional Chinese medicine for treating pulmonary fibrosis, was given for 7 or 28 days to suppress M2 macrophage polarization [57,59]. The results showed that suppression of M2 polarization by Schisandra was associated with the inhibition of bleomycin-induced pulmonary fibrosis. Likewise, Wang et al. reported the inhibition of CD206+ M2 polarization of macrophages, using microcystinleucine arginine, which eventually alleviated pulmonary fibrosis [60].

Currently, increasing evidence suggests that EVs derived from macrophages are also involved in regulating pulmonary fibrosis. A literature search of all preclinical studies on the effect of vesicles derived from macrophages on pulmonary fibrosis was conducted, according to the approach described above, using the keywords “EVs”, “extracellular vesicles” or “exosomes” and “macrophage”, together with “pulmonary fibrosis”. A total of four articles were identified [61,62,63,64]; of these studies, three indicated a profibrogenic effect, while one revealed a suppressive effect. Specifically, Yao et al. illustrated that miR-328-containing EVs derived from M2 macrophages stimulated pulmonary fibrosis in a rat model [61]. Mechanistically, miR-328 might exert a vital function by regulating FAM13A. FAM13A expression was downregulated when the miR-328 expression was upregulated. Moreover, a miR-target relationship between miR-328 and FAM13A was identified. Silencing of FAM13A promoted pulmonary interstitial fibroblast proliferation and the expression of Collagen 1A, Collagen 3A, and α-SMA. Likewise, silica-exposed macrophage-derived EVs were collected and cocultured with fibroblasts, revealing reduced expression of collagen I and α-SMA. However, mice pretreated with the EV-secretion inhibitor GW4869 prior to silica exposure showed decreased lung fibrosis and expression of TNF-α, IL-1β, and IL-6 [62]. In another study, Yamada et al. [65] investigated the quantity of miRNAs in serum EVs of mice with bleomycin-induced lung fibrosis, and reported significant upregulation of serum EV-miR-21e5p in the acute and chronic fibrotic phases. Furthermore, as miR-21e5p promotes TGF-b signaling, a critical signaling pathway in pulmonary fibrosis, miR-21e5p was suggested to be a potential biomarker of pulmonary fibrosis. Guiot et al. revealed that sputum macrophages in pulmonary fibrosis were found to contain elevated levels of EV-miR-142-3p, and macrophage-derived EVs could fight against pulmonary fibrosis progression by transferring antifibrotic miR-142–3p to alveolar epithelial cells and lung fibroblasts. By characterizing the miRNA content of sputum EVs of patients with pulmonary fibrosis versus healthy subjects, miR-142-3p was identified as a novel diagnostic biomarker [66]. It should be noted that sample sizes in preclinical and clinical experiments have so far been insufficient, indicating that further high-quality experiments are required [63]. More information regarding macrophage EVs’ effects on the pathogenesis of pulmonary fibrosis is summarized in Table 2.

## 6. Macrophages Play an Important Role for Mesenchymal Stem Cells to Exert Protective Effects

MSC-based and macrophage-based cell therapies are deemed promising strategies to improve fracture healing, due to the incredible protective potential of MSCs and the typical immunomodulatory effects of macrophages. Macrophages are vital for MSCs to exert their protective effects; for instance, depletion of macrophages by lipoclodronate solution represses the protective function of MSC [67]. MSCs or MSC-EVs serve as crucial mediators to modulate macrophage behavior according to two major patterns, including attenuating macrophage activation and M1 phenotype macrophage polarization [67]. The transplantation of MSCs into rats three hours after focal cerebral ischemia, for example, showed a significant reduction in macrophages on day three after treatment [68]. Concerning macrophage polarization, according to reports by Abumaree et al. [69] and Maggini et al. [70], MSCs can suppress M1 markers in vitro, namely TNF-α and iNOS; meanwhile, they promote the differentiation of macrophages toward the M2 phenotype, producing IL-10, CD206, and Arg1. Further evidence provided by Dayan et al. [71] from observations of a myocardial infarction condition revealed that, compared with the non-MSC group, the MSC group was associated with higher levels of Arg1 and IL-10 and lower expression of proinflammatory M1 markers of IL-1β and IL-6. This observation is consistent with similar reports on MSC-EVs, in which tissue restoration was boosted due to the inhibition of M1 macrophage polarization after MSC-EVs treatment [72,73,74,75]. Interestingly, emerging evidence has demonstrated that MSC-EVs can also modulate macrophage polarization in pulmonary fibrosis, contributing to lung repair. Wang et al. [60] demonstrated that MSC-EVs mitigated pulmonary fibrosis at least partially by transferring their cargos to macrophages, further promoting M2 macrophage polarization and eventually contributing to the alleviation of pulmonary fibrosis. Notably, protective effects against pulmonary fibrosis differ between young and aging MSC-EVs, due to different effects on the modulation of M2 macrophage polarization. As reported by Huang et al. [54], compared to young MSC-EVs, aging MSC-EVs showed impaired therapeutic effects in a murine model of LPS-induced acute lung injury, and were deficient in promoting M2 macrophage polarization and suppressing macrophage activation. Mechanistically, aging MSC-EVs are less readily internalized by macrophages compared with their younger counterparts. MSC-EVs were reported to mitigate pulmonary fibrosis at least partially by inhibiting macrophage recruitment and promoting M2 macrophage polarization.

## 7. Extracellular Vesicle-Associated miRNA as One of the Dominating Effectors

### 7.1. miRNA: Biogenesis and Characteristics

With the revolution in techniques for gene sequencing, the transcribed genome encodes over 20,000 types of protein; however, only ~2% of the whole human genome encodes for proteins [76,77]. Not all RNAs can translate functional proteins, and RNAs can be classified into two main subcategories—those with the function of coding and those without, known as ncRNAs [78]. Nevertheless, emerging evidence demonstrates that ncRNAs play a vital role in the modulation of gene expression and contribute to numerous disorders [78,79]. Based on their nucleotide length, ncRNAs are commonly classified into long (namely lncRNA and circRNA) and small ncRNAs (miRNAs, tRNAs, piRNAs), taking 200 nucleotides as the limit [80]. miRNAs, one kind of endogenous ncRNAs, were reported much earlier (first found in Caenorhabditis elegans in 1993) and are the best-described type of small ncRNAs, no more than 18–24 nucleotides in length [81,82]. In vertebrates, a total of five stages are involved in the biogenesis of miRNA: (I) Type-II RNA polymerases promote the transcription of miRNAs from DNA facilitated into pre-miRNAs [83,84]; (II) pri-miRNAs are processed by various microprocessor complexes [84,85]; (III) Ran-GTP and Exportin-5 help the pre-miRNA export from the nucleus [86,87]; (IV) the pre-miRNA is further cleaved and forms a miRNA duplex by binding to the RNA-binding proteins; and (V) the RISC is incorporated into an RNP that assimilates the mature miRNA [88,89]. Despite the fact that less than 0.02% of the cells’ total RNA content consists of miRNAs [78,90], it is well established that miRNAs are involved in regulating approximately 60% of all protein-coding genes, and miRNAs have an average of 200 targets [91,92]. miRNA is a family of post-transcriptional gene repressors that modulate gene expression by combining with the 3’ untranslated region of the target mRNA sequence, inhibiting mRNA levels [93].

### 7.2. miRNA Loading into EVs and Uptake by Recipient Cells

As described above, a wide array of molecules, including proteins, lipids, DNAs, mRNAs, and miRNAs, can be transferred from donor to recipient cells by EVs from different cell types [94,95,96]. Among these molecules, exosomal miRNAs have attracted increasing interest. Although the underlying mechanisms are still not entirely understood, four proposed modes for loading and sorting miRNAs into EVs have been confirmed [97]. These pathways include the sphingomyelinase 2-dependent pathway, sumoylated hnRNP-dependent pathway (mainly including hnRNPA2B1, hnRNPA1, and hnRNPC), miRNA-induced silencing complex (miRISC)-associated pathway, the RNA-binding protein pathway, and the 3’ end of the miRNA sequence-dependent pathway [98,99,100,101]. It is generally accepted that EVs’ size, surface components, and physical characteristics probably influence their recognition and capture by target cells [102]. Once released to the extracellular environment, the exosomes can enter cells and interact with recipient cells to transmit information by several different mechanisms. The widely discussed EV uptake mechanisms include direct fusion, receptor-mediated fusion, and endocytosis [103,104]. There have also been reports of direct cell-to-cell extracellular vesicle transfer via tunneling nanotubes for RNA delivery [105]. Typically, the EVs attach to the recipient cells and release their content into the target cells by fusing with the plasma membrane or entering the cell [106]. Following internalization, various molecular cargoes in EVs can be released into the cytoplasm, delivered to lysosomes, destroyed, or targeted to specific locations within the cell [107]. However, it remains unresolved whether the cells respond to specific EVs or whether the process is unspecific and stochastic. After being delivered to the target cells, the extracellular miRNAs can exert the translation of the target genes and the function of the target cells, thereby acting as intercellular signaling molecules [108]. The brief mechanism of how miRNA loading into EV and uptake by recipient cells was exhibited in Figure 1.

The maturation of the miRNA form mainly by the canonical pathway. Based on the EVs’ sizes, they can be divided into three subtypes: exosomes, microvesicles, and apoptotic bodies. miRNAs can be loaded into extracellular vesicles via various RNA-binding proteins. These extracellular vesicles containing miRNAs are released into the extracellular space. Recipient cells can uptake exosomal miRNAs through direct fusion, endocytosis, and receptor signaling.

### 7.3. EV-miRNA Is One of the Dominating Effectors

Although fibrotic interstitial lung disease limits the lesion site in the lung, it results in a series of pathological events, among which fibroblast proliferation and differentiation, apoptosis, autophagy, and inflammatory damage may ultimately lead to lung function failure [109]. Due perhaps to the chronic and progressive nature and severity of lung injury induced by pulmonary fibrosis, several profibrotic miRNAs are downregulated in lung tissue in response to pulmonary fibrosis, as previously shown for miR-21, miR-506, miR-96, miR-499a, miR-326, miR-410, miR-124, miR-328, miR-420, miR-7, miR-19a, miR-19b, miR-26b, miR-9, miR-29, and miR-30a [110,111]. In contrast, other antifibrotic miRNAs, such as miR-30, miR-101, miR-344, miR-323a-3p, miR-29b, miR-185, miR-29a, miR-185, miR-186, miR-221, miR-1343, miR-27a-3p, and miR-27b, are increased [110,111]. miRNAs have been shown to play an essential role in these pathological events. For example, several miRNAs, namely miR-328, miR-420, miR-7, miR-19a, miR-19b, and miR-26b, can ameliorate pulmonary fibrosis by inhibiting fibroblast differentiation or proliferation, whereas other miRNA molecules, such as miR-30, miR-101, and miR-344 exert the opposite effect [110,112,113]. miR-30a and miR-29 have been shown to be significantly downregulated in a murine model of lung fibrosis [114,115], promoting epithelial cell apoptosis resistance. However, miR-34a upregulation was found to accelerate lung epithelial cell apoptosis, causing epithelial cell dysfunction and increased lung fibrosis [116]. The overexpression of miR-499a and miR-326 significantly reduced lung fibrosis and promoted autophagy in vivo and in vitro [117,118]. Additionally, miR-96 silencing can result in upregulation of FOXO3a, thereby promoting the activation of NLRP3 inflammasome [119].

As mentioned above, macrophages are vital cells in the immune response; however, they can also promote pulmonary fibrosis. EV-miRNAs derived from macrophages are also involved in mediating pulmonary fibrosis. For example, the overexpression of M2 macrophage-derived EV-miR-328 contributed to boosted fibroblast proliferation and increased pulmonary fibrosis by modulating FAM13A [61]. Another study using high-throughput sequencing examined EV-miRNA profiles from macrophages exposed to silica. Compared with unexposed macrophages, 155 miRNAs were upregulated, and 143 miRNAs were downregulated [120]. Delivery of EVs derived from various MSCs appeared to restore reduced levels of miRNA in fibrotic lung tissue. EV-miRNAs play important roles in coordinating responses to pulmonary fibrosis by modulating various pathological processes. miR-186, miR-29b-3p, miR-21-5p, miR-182-5p, and miR-23a-3p from bone marrow MSC-derived EVs [38,40,121,122], miR-27a-3p from adipose-derived EVs [53], and miR-Let-7 from menstrual blood-derived endometrial stem cell-associated EVs have been shown to inhibit pulmonary fibrosis [50]. Specifically, miR-29b-3p and miR-186 have the potential to inhibit the activation of fibroblasts by regulating FZD6, SOX4, and DKK1. miR-21-5p, miR-182-5p, miR-23a-3p, miR-Let-7, and miR-27a-3p can reduce the synthesis of proinflammatory cytokines. They can also inhibit inflammation-activation-associated signaling pathways (such as NLRP3 inflammasome and NF-κB). Notably, miRNAs from MSC-EVs can also modulate macrophage polarization in pulmonary fibrosis. Compared with aging MSC-EVs, young MSC-EVs are related to lower levels of miR-127-3p and miR-125b-5p. Furthermore, inhibition of miR-127-3p and miR-125b-5p in bone-marrow-derived macrophages was reported to downregulate M1 macrophage polarization [54]. Likewise, miR-27a-3p was reported to be the key effector of MSC-EVs in mitigating pulmonary fibrosis. Mechanistically, miR-27a-3p was revealed to promote M2 macrophage polarization by targeting NFKB1 [60]. Figure 2 described more details.

Mesenchymal stem cells are isolated and identified from various tissue sources. Mesenchymal stem cells can produce various EV-miRNAs, which contribute to inhibiting pulmonary fibrosis, whereas EV-miRNAs derived from macrophages can promote pulmonary fibrosis. Of note, miRNAs from MSC-EVs can also modulate macrophage activation and polarization, suppressing pulmonary fibrosis.

## 8. Conclusions

MSC-based and macrophage-based cell therapies are deemed promising strategies to promote lung repair, due to tremendous regenerative potential and typical immunomodulatory function. EVs are lipid bilayer-covered nanoparticles secreted from virtually all cell types, and are a significant component of the broader class of nanoparticles that have essential roles in intercellular communication. In this paper, we have shown that EVs from MSCs and macrophages are profoundly involved in controlling the physiological responses of pulmonary fibrosis. Macrophages are vital for MSCs to exert their protective effects by attenuating macrophage activation and M1 phenotype macrophage polarization. As essential components and effectors of EVs, miRNAs selectively sorted into EVs potentially regulate specific aspects of the onset and progression of pulmonary fibrosis. However, the study of EV miRNAs from MSCs and macrophages in pulmonary fibrosis remains in its infancy, and further research is required to investigate their role.

## Figures and Tables

**Figure 1 pharmaceuticals-15-01276-f001:**
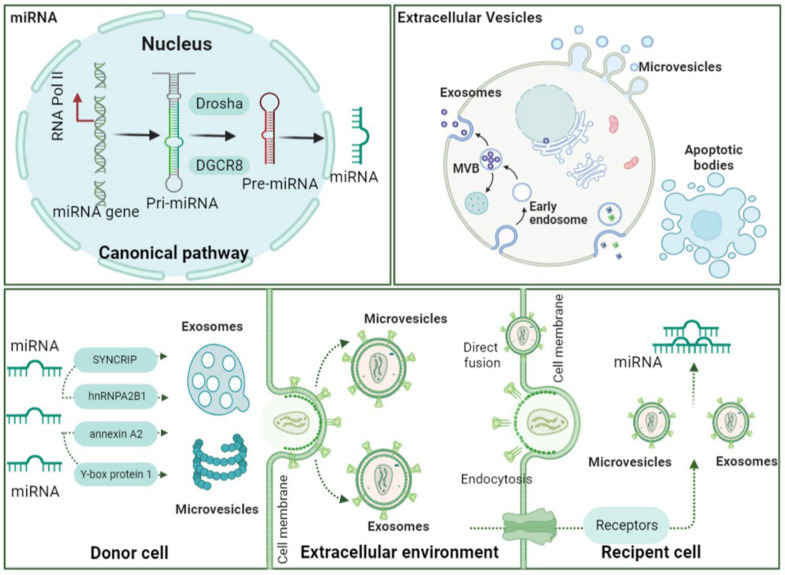
The brief mechanism of how miRNA loading into EV and uptake by recipient cells.

**Figure 2 pharmaceuticals-15-01276-f002:**
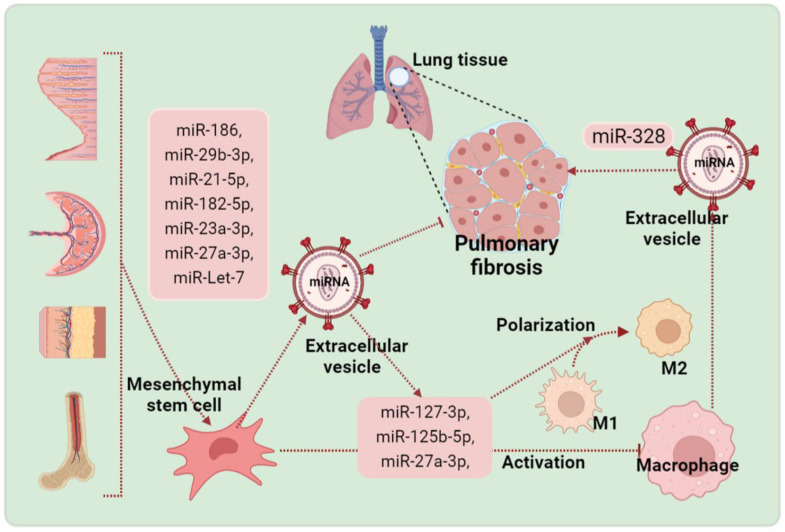
The underlying therapeutic action of extracellular vesicle miRNA from mesenchymal stem cells in pulmonary fibrosis.

**Table 1 pharmaceuticals-15-01276-t001:** Summary of studies in animal models evaluating the effect of mesenchymal stem cell extracellular vesicles in pulmonary fibrosis.

Author, Year	Country	EVs Source	Dosage	Administration	Primary Effects
**Bandeira et al. [47] 2018**	Brazil	AD-MSC	EVs from 10^5^ MSCs	Intratracheal injection	Ameliorates fibrosis and inflammation
**Xu et al. [44] 2022**	China	Huc-MSC	Not available	Not available	Transfers let-7i-5p to inhibit pulmonary fibrosis
**Xu et al. [45] 2020**	China	Huc-MSC	100 µg/250 µL	Tail vein injection	Inhibits silica-induced pulmonary fibrosis and regulate the pulmonary function
**Li et al. [46] 2021**	China	Huc-MSC	20 µg	Tail vein injection	Alleviates pulmonary fibrosis and enhance the proliferation of alveolar epithelial cells
**Tan et al. [48] 2018**	Australia	Amnion Epithelial Cell	10 µg	Intranasal administration	Demonstrates potent antifibrotic, immunomodulatory, and regenerative properties
**Yang et al. [49] 2022**	China	Embryonic MSC	200 µg or 1000 µg	Intratracheal or tail vein injection	Inhibits bleomycin-induced pulmonary fibrosis
**Sun et al. [50] 2019**	China	Human MSC	0.5 mg/kg/day	Tail vein injection	Causes remittance of pulmonary fibrosis by regulating ROS, mtDNA damage, and NLRP3 inflammasome activation
**Lei et al. [51] 2021**	China	Placenta MSC	100 µg	Tail vein injection	Attenuates radiation-induced lung injury via miRNA-214-3p
**Kusuma et al. [36] 2022**	Australia	Human BMSC	10 µg	Intranasal administration	Exhibits immunomodulation, antifibrotic, and anti-inflammatory effects
**Zhang et al. [37] 2021**	China	Rat BMSC	Tail vein injection	200 µg/mL/rat	Reverses epithelial-mesenchymal transition via Wnt/β-catenin to alleviate silica-induced pulmonary fibrosis
**Xiao et al. [38] 2020**	China	Human BMSC	Tail vein injection	70 µg	Reverses EMT process via blocking NF-κB and Hedgehog in LPS-induced acute lung injury
**Mansouri et al. [39] 2019**	USA	Human BMSC	Tail vein injection	200 µL, 8.6 × 10^8^ particles	Prevents and reverts experimental pulmonary fibrosis through modulation of monocyte phenotypes
**Wan et al. [40] 2020**	China	Human BMSC	Tail vein injection	100 µg	Suppresses pulmonary fibrosis by downregulating FZD6 in fibroblasts via microRNA-29b-3p
**Zhou et al. [41] 2021**	China	Human BMSC	Tail vein injection	100 µg	Alleviates pulmonary fibrosis via interaction with SOX4 and DKK1
**Li et al. [52] 2022**	China	Mouse MSC	Tail vein injection	200 µg	Reverses EMT process by inhibiting AKT/GSK3β pathway via c-MET in radiation-induced lung injury
**Choi et al. [42] 2014**	Korea	Human BMSC	Tail vein injection	10 µg	Exerts a cytoprotective effect on reducing pulmonary fibrosis, such as collagen deposition and inflammation
**Rozier et al. [43] 2021**	France	Mouse MSC	Intravenous injection	250 ng or 1500 ng	Improves lung repair by modulating anti-inflammatory and antifibrotic markers

**Note:** EVs, extracellular vesicles; MSCs, mesenchymal stem cells; BMSC, bone marrow mesenchymal stem cells; Huc-MSC, human umbilical cord mesenchymal stem cells; AD-MSC, adipose mesenchymal stem cells; Men MSC, menstrual blood-derived endometrial stem cells.

**Table 2 pharmaceuticals-15-01276-t002:** Summary of studies in animal models evaluating the effect of macrophage extracellular vesicles in pulmonary fibrosis.

Author, Year	Country	EVs Source	Dosage	Administration	Primary Effects
**Yao et al. [61] 2019**	China	M2 macrophages	100 μg	Tail vein injection	Exerts a promotive effect on the progression of pulmonary fibrosis via FAM13A
**Qin et al. [62] 2021**	China	Silica-exposed macrophage	50 μg	Not available	EVs are profibrogenic and contribute to pulmonary fibrosis and inflammation during silicosis
**Guiot et al. [63] 2020**	Belgium	Macrophage	Not available	Not available	Prevents pulmonary fibrosis progression via the delivery of miR-142–3 p to alveolar epithelial cells and lung fibroblasts
**Sun et al. [64] 2021**	China	Macrophage	1 g/kg	Tail vein injection	Transfers angiotensin II type 1 receptor to lung fibroblasts mediating bleomycin-induced pulmonary fibrosis

**Note:** EVs, extracellular vesicles.

## Data Availability

Data sharing not applicable.

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
