# Peer review of "The Emerging Role of Extracellular Vesicles from Mesenchymal Stem Cells and Macrophages in Pulmonary Fibrosis: Insights into miRNA Delivery"

_pharmaceuticals, 2022, doi:10.3390/ph15101276_

Round 1

Reviewer 1 Report

In this review, the authors stated that MSC-based and macrophage-based cell therapy is deemed a promising strategy to promote lung repair due to its incredible regenerative potential and typical immunomodulatory function. Extracellular vesicles (EVs), which include exosomes and microvesicles, are a class of cell-derived lipid-bilayer membrane vesicles secreted from virtually every cell and involved in intercellular communication by delivering biological cargos to recipients. This review takes a deep insight into the recent research progress concerning the effect of MSC and macrophage-associated EVs on the pathogenesis of pulmonary fibrosis and summarizes the importance of cross-talk since macrophages are vital for MSCs to exert their protective effects by two major patterns, including attenuating macrophage activation and M1 phenotype macrophage polarization. In addition, miRNAs were selectively enriched into EVs, and this review illustrates the effect of EVs-associated miRNAs on pulmonary fibrosis.  In general this is an ambitious review, it tried to cover two typical cell therapy (MSC and macrophage) on pulmonary fibrosis, and establish connection between cells and their EVs. However, the topic is huge, and it lacks latest details to explain the effects and mechanisms. 

The reviewer suggested the author to consider focusing on one topic, for example, EVs derived from mesenchymal stem cells (MSCs) in the pathogenesis of pulmonary fibrosis, or EVs derived from macrophages in the pathogenesis of pulmonaryfibrosis, or the interaction of mesenchymal stem cells and macrophages (or the message deliver by MSC’s EV to macrophages?).  A lot of things were listed in this review and no clear logic clue could be followed.  

Major suggestions:

1.    In introduction, the authors mentioned “Mesenchymal stem cells (MSCs) transplantation is attractive therapy since they have the potential to self-renew and exhibit multilineage differentiation and immunomodulatory properties”. However, it was frequently observed that functional improvement after MSC transplantation does not correlate with engraftment or differentiation of MSCs. Some observations have hypothesis that MSCs reduced injury and repaired tissues through their secretion factors (EV is just part of that). This should be explained in detail.

2.    In part “Umbilical cord MSC-derived EVs”, the authors should illustrate how EVs from three-dimensional cultured umbilical cord MSC had the potential to repress silica-induced pulmonary fibrosis and improve lung function.

3.    In part “Other MSC-derived EVs”, the authors mentioned “Human embryonic stem cell EVs administrated showed a therapeutic effect as attenuated pulmonary fibrosis, improved lung function, and histological parameters”. The authors should illustrate the mechanism of EVs in reducing pulmonary fibrosis and improving pulmonary function.

4.    In part “The interaction of mesenchymal stem cells and macrophages in the pathogenesis of

pulmonary fibrosis”, the authors mentioned “MSCs or MSC-EVs serve as a crucial mediator to modulate macrophage behavior by two major patterns, including attenuating macrophage activation and M1 phenotype macrophage polarization”. The authors should clarify whether MSC is different from MSCS-EVs efficacy on macrophages. Do MSCs exert interactions with macrophages through MSCs-EVs or MSCs have direct contact with macrophages?

5.    In part “EV-miRNAs are the dominating effectors”, the author thinks that miR-7, miR-29, and miR-30a are profibrotic miRNAs.  To the reviewer’s understanding, the roles are opposite.  Besides, in part 7.3, the arguments provided were insufficient to support the conclusion that EV-miRNAs are the dominating effectors. Exosomes have multiple pharmacodynamic effects through different ways, miRNA is just one of them.

Minor suggestions:

1.  In introduction, the authors mentioned “Cell therapy is emerging as a promising novel modality for facilitating lung repair. Mesenchymal stem cells (MSCs) transplantation is attractive therapy since they have the potential to self-renew and exhibit multilineage differentiation and immuno- modulatory properties”. However, this review focuses on the antifibrotic effect of EVs. The advantages of EVs compared with cell therapy need to be addressed.

2.    In part “EVs from adipose-tissue-derived MSCs”, the authors mentioned “Interestingly, adipose-tissue-derived EVs can be efficiently taken up by alveolar macrophages in vitro and in vivo”. It should be clear whether " adipose-derived EVs " is EVs from adipose-tissue-derived MSC.

Author Response

Xiuru Zhang, PhD, MD, MSc

Professor of Pathology, Department of Pathology

The Third People's Hospital of Longgang District

278 Songbai Road, Henggang Street, Shenzhen, China

Email: zhangxiurudr@126.com; Phone: +86–15350402147

Eden Wang, Ph.D.

Assistant Editor, Pharmaceuticals

Pharmaceuticals-1918912 “The emerging role of extracellular vesicles from mesenchymal stem cells and macrophages in pulmonary fibrosis: Insights into miRNAs delivery”

Dear Dr. Wang,

Thank you very much for sending us the reviews of our manuscript. The comments were all valuable and were very helpful for revising and improving our paper. According to the reviewers' and editor's comments and suggestions, we have revised the manuscript and responded, point by point, to the comments, as listed below. Revised portions are marked in light grey on the paper. All authors have read and approved the final version of the manuscript. We would like to thank you for considering our work for publication in Pharmaceuticals.

Yours sincerely,

Prof. Xiuru Zhang, PhD, MD, MSc.

(On behalf of all co-authors)

Replies to Reviewers and Editor

First, we thank both reviewers and the editor again for their careful review and valuable comments. The comments are all helpful for revising and improving our manuscript.

Reviewer 1’s Comments

* Reviewer1 #1: Comments to the Author

*1. In this review, the authors stated that MSC-based and macrophage-based cell therapy is deemed a promising strategy to promote lung repair due to its incredible regenerative potential and typical immunomodulatory function. Extracellular vesicles (EVs), which include exosomes and microvesicles, are a class of cell-derived lipid-bilayer membrane vesicles secreted from virtually every cell and involved in intercellular communication by delivering biological cargos to recipients. This review takes a deep insight into the recent research progress concerning the effect of MSC and macrophage-associated EVs on the pathogenesis of pulmonary fibrosis and summarizes the importance of cross-talk since macrophages are vital for MSCs to exert their protective effects by two major patterns, including attenuating macrophage activation and M1 phenotype macrophage polarization. In addition, miRNAs were selectively enriched into EVs, and this review illustrates the effect of EVs-associated miRNAs on pulmonary fibrosis. In general, this is an ambitious review, it tried to cover two typical cell therapy (MSC and macrophage) on pulmonary fibrosis, and establish connection between cells and their EVs. However, the topic is huge, and it lacks latest details to explain the effects and mechanisms.

Answer: Many thanks for your careful review and positive comments. I couldn't agree more with you the topic is huge. We conducted a systematic review before this review, a total of 17 studies uncovered the effects of MSC-EVs on the treatment of pulmonary fibrosis. Only 4 studies uncovered the effects of macrophage -EVs on the treatment of pulmonary fibrosis. Afterward, we believe that the current information is insufficient to support us to separate them into multiple topics. Therefore, this review takes a deep insight into the recent research progress concerning the effect of MSC and macrophage-associated EVs on the treatment of pulmonary fibrosis. Moreover, a notion is given to the importance of cross-talk since macrophages are vital for MSCs to exert their protective effects. Regarding latest details to explain the effects and mechanisms. More information is summarized in Table 1 and Table 2.

* Reviewer1 #2: Comments to the Author

*2. The reviewer suggested the author to consider focusing on one topic, for example, EVs derived from mesenchymal stem cells (MSCs) in the pathogenesis of pulmonary fibrosis, or EVs derived from macrophages in the pathogenesis of pulmonary fibrosis, or the interaction of mesenchymal stem cells and macrophages (or the message deliver by MSC’s EV to macrophages?). A lot of things were listed in this review and no clear logic clue could be followed.

Answer: Many thanks for your careful review and positive comments. Thanks again. I couldn't agree more with you the topic is huge. We conducted a systematic review before this review, a total of 17 studies uncovered the effects of MSC-EVs on the treatment of pulmonary fibrosis. Only 4 studies uncovered the effects of macrophage -EVs on the treatment of pulmonary fibrosis. Afterward, we believe that the current information is insufficient to support us to separate them into multiple topics. Therefore, this review takes a deep insight into the recent research progress concerning the effect of MSC and macrophage-associated EVs on the treatment of pulmonary fibrosis. Moreover, a notion is given to the importance of cross-talk since macrophages are vital for MSCs to exert their protective effects.

* Reviewer1 #3: Comments to the Author

*3. Major suggestions: In introduction, the authors mentioned “Mesenchymal stem cells (MSCs) transplantation is attractive therapy since they have the potential to self-renew and exhibit multilineage differentiation and immunomodulatory properties”. However, it was frequently observed that functional improvement after MSC transplantation does not correlate with engraftment or differentiation of MSCs. Some observations have hypothesis that MSCs reduced injury and repaired tissues through their secretion factors (EV is just part of that). This should be explained in detail.

Answer: Many thanks for your careful review and positive comments. I couldn't agree more with you, however, we must admit that the reason why people first studied stem cells since they have the potential to self-renew and exhibit multilineage differentiation and immunomodulatory properties. With the development of later studies, MSCs often exercise their protective functions with the help of EVs secreted by them. Therefore, we have revised it as your suggestion in the section of introduction. (Line 48-50, Line 59-64, Line 66-72, Line 73-76)

* Reviewer1 #4: Comments to the Author

*4. In part “Umbilical cord MSC-derived EVs”, the authors should illustrate how EVs from three-dimensional cultured umbilical cord MSC had the potential to repress silica-induced pulmonary fibrosis and improve lung function.

Answer: Many thanks for your careful review and positive comments. We have revised it as your suggestions. (Line 154-166)

* Reviewer1 #5: Comments to the Author

*5. In part “Other MSC-derived EVs”, the authors mentioned “Human embryonic stem cell EVs administrated showed a therapeutic effect as attenuated pulmonary fibrosis, improved lung function, and histological parameters”. The authors should illustrate the mechanism of EVs in reducing pulmonary fibrosis and improving pulmonary function.

Answer: Many thanks for your careful review and positive comments. We have added some information regarding the mechanism of EVs in reducing pulmonary fibrosis and improving pulmonary function as your suggestions. (Line 192-196)

* Reviewer1 #6: Comments to the Author

*6. In part “The interaction of mesenchymal stem cells and macrophages in the pathogenesis of pulmonary fibrosis”, the authors mentioned “MSCs or MSC-EVs serve as a crucial mediator to modulate macrophage behavior by two major patterns, including attenuating macrophage activation and M1 phenotype macrophage polarization”. The authors should clarify whether MSC is different from MSCS-EVs efficacy on macrophages. Do MSCs exert interactions with macrophages through MSCs-EVs or MSCs have direct contact with macrophages?

Answer: Many thanks for your careful review and positive comments. I could not agree with you more that it would be interesting to clarify whether MSC is different from MSCS-EVs efficacy on macrophages. However, there is no direct evidence showing the different effects between the MSC and MSC-EVs in terms of the treatment of pulmonary fibrosis. Current evidence shows that both MCS and MSC-EVs have regulatory effects on macrophages, thereby we can think that the effect of MSC on macrophages is, at least partly, through its way of secreting EVs.

* Reviewer1 #7: Comments to the Author

*7. In part “EV-miRNAs are the dominating effectors”, the author thinks that miR-7, miR-29, and miR-30a are profibrotic miRNAs.  To the reviewer’s understanding, the roles are opposite.  Besides, in part 7.3, the arguments provided were insufficient to support the conclusion that EV-miRNAs are the dominating effectors. Exosomes have multiple pharmacodynamic effects through different ways, miRNA is just one of them.

Answer: Many thanks for your careful review and positive comments. I think there may be an understanding deviation, we also think that miR-29 and miR-30a are antifibrotic miRNAs. As described in the manuscript, “miR‑30a and miR‑29 have been uncovered to be significantly downregulated in a murine model of lung fibrosis, which promote epithelial cell apoptosis resistance, whereas miR‑34a upregulation has been found to accelerate lung epithelial cell apoptosis, causing epithelial cell dysfunction and increased lung fibrosis.”

Meanwhile, we agree that EVs have multiple pharmacodynamic effects through different ways, miRNA is just one of them. Therefore, we have revised them into “EV-miRNA is one of the dominating effectors”. (Line 293, Line-344)

 * Reviewer1 #8: Comments to the Author

*8. Minor suggestions: In introduction, the authors mentioned “Cell therapy is emerging as a promising novel modality for facilitating lung repair. Mesenchymal stem cells (MSCs) transplantation is attractive therapy since they have the potential to self-renew and exhibit multilineage differentiation and immuno- modulatory properties”. However, this review focuses on the antifibrotic effect of EVs. The advantages of EVs compared with cell therapy need to be addressed.

Answer: Many thanks for your careful review and positive comments. We have added some information regarding this issue. (Line 48-50, Line 59-64, Line 66-72, Line 73-76)

* Reviewer1 #9: Comments to the Author

*9. In part “EVs from adipose-tissue-derived MSCs”, the authors mentioned “Interestingly, adipose-tissue-derived EVs can be efficiently taken up by alveolar macrophages in vitro and in vivo”. It should be clear whether " adipose-derived EVs " is EVs from adipose-tissue-derived MSC.

Answer: Many thanks for your careful review and positive comments. It should be “adipose-derived EVs”. We have revised them totally as your suggestions. (Line 133-152)

Reviewer 2’s Comments

* Reviewer2 #1: Comments to the Author

*1. This manuscript reviews the current knowledge in the field of EVs for pulmonary fibrosis and specifically the specific miRNA in EVs that are known to contribute to/ alleviate pulmonary fibrosis. To improve the manuscript, I have included specific comments below.

Abstract: Line 20 – Not sure if ‘prototype’ is the best word to use here

Answer: Many thanks for your careful review and positive comments. We have changed the ‘prototype’ into “type” (Line 20)

* Reviewer2 #2: Comments to the Author

*2. Introduction - Gives a very general overview of pulmonary fibrosis (PF), more is required on the current gaps in knowledge and how this review fulfils the needs in the field.

Answer: Many thanks for your careful review, positive comments, and affirming our research. We have added some information in the section of introduction as all reviewers’ suggestions. (Line 48-50, Line 59-64, Line 66-72, Line 73-76)

* Reviewer2 #3: Comments to the Author

*3. Line 63 – should mention that EVs are classed not only size but also other factors, e.g., their biogenesis, specific markers, function.

Answer: Many thanks for your careful review and positive comments. We have revised it as your suggestion. (Line 68-72)

* Reviewer2 #4: Comments to the Author

*4. Line 74 – ‘EVs as therapeutic targets for PF’? This is not a correct description as the manuscript is more on EVs as therapeutic ‘approaches’ for PF.

Answer: Many thanks for your careful review and positive comments. We have change the " targets" into “tools” (Line 84-85)

* Reviewer2 #5: Comments to the Author

*5. Line 64, 96 – Apoptotic bodies can be much smaller than 800nm – this should be corrected

Answer: Many thanks for your careful review and positive comments. We have revised them as your suggestions. (Line 68, Line 106)

* Reviewer2 #6: Comments to the Author

*6. Heading for Section 4 – ‘Contributions of EVs derived from MSCs in the pathogenesis of pulmonary fibrosis’ – should ‘pathogeneses” be replaced with ‘treatment’?

Answer: Many thanks for your careful review and positive comments. We have changed ‘pathogeneses” into “treatment” as your suggestions. (Line 116-117)

* Reviewer2 #7: Comments to the Author

*7. Line 127 – the sentence starting with ‘Upon an acute lung injury condition.’ should be modified to ensure that correct message is conveyed – there are many causes of PF, often unknown, but acute lung injury (should of course specify the specific types of acute lung injury) can contribute to the development of PF.

Answer: Many thanks for your careful review and positive comments. We have revised it as your suggestions. (Line 137-138)

* Reviewer2 #8: Comments to the Author

*8. Line 173-184 – this section titled ‘Other MSC-derived EVs’ contain lots of useful information regarding how EVs can alter the course of PF. This is the section that differentiates this manuscript from a large volume of other papers that superficially mention how EVs can be used for the treatment of PF. Please could you expand more on how these EVs and the specific miRNAs named in this paper are used in what aspect of PF pathophysiology. Include what the authors’ views are in using these specific types of EVs. Then this review will be able to help future researchers working on new approaches to target PF and fulfil the purpose of this review ‘particular emphasis on the effects of EV-associated miRNAs’ as stated in line 74-75.

Answer: Many thanks for your careful review and positive comments. We have expanded this section as your suggestions. (Line 192-200)

* Reviewer2 #9: Comments to the Author

*9. Section 5 is a critical part of this manuscript as it adds to the field much more than MSC-derived EVs (since there is a huge number of paper and knowledge on this). This section should be expanded further to include more details on the profibrogenic effects of EVs – e.g., what are the specific roles of the miR-328 inside EVs? Are there any other studies that report the role of the antifibrotic miR-142-3p in the field of PF?

Answer: Many thanks for your careful review and positive comments. We have expanded this section as your suggestions. (Line 234-240, Line 244-254)

* Reviewer2 #10: Comments to the Author

*10. Line 205 – include a brief description of Schisandra when it’s first introduced.

Answer: Many thanks for your careful review and positive comments. We have added a a brief description of Schisandra as your suggestions. (Line 221-223)

* Reviewer2 #11: Comments to the Author

*11. Line 213 – ‘extracellular vehicles’ should be corrected to ‘vesicles’

Answer: Many thanks for your careful review and positive comments. We have revised it as your suggestions. (Line 230)

* Reviewer2 #12: Comments to the Author

*12. The title of Section 6 should be modified to better reflect the text/content in that section

Answer: Many thanks for your careful review and positive comments. We have changed the title of Section 6 into “Macrophages play an important role for mesenchymal stem cells to exert protective effects” (Line 259-260)

* Reviewer2 #13: Comments to the Author

*13. Section 7 is the highlight of this manuscript, containing key information that differentiate this paper from other published papers. Consider moving Section 7 up towards the beginning of the manuscript, as it was quite difficult to wait until the end of the manuscript to find the content I anticipated from the title of the manuscript.

Answer: Many thanks for your careful review and positive comments. I am very agreed with you, but since the current article is a progressive relationship, if we rashly put this part of the content in the front, it will completely disrupt the logic of the article. We currently want to keep this section in its current location. Thanks again.

* Reviewer2 #14: Comments to the Author

*14. Line 344- when these EV-miRNAs play important roles in PF, are they eventually contributing to the pro-fibrotic processes or anti-fibrotic processes? Are these miRNAs the same as those found in serum/blood of PF patients?

Answer: Many thanks for your careful review and positive comments. Line 344: “Another study, adopting high-throughput sequencing, examined EV-miRNA profiles from macrophages exposed to silica; compared with unexposed macrophages, 155 miRNAs were upregulated, and 143 miRNAs were downregulated”

We want to say that the miRNA has changed and these may play important roles in lung fibrosis. Additionally, the original study did not show these miRNAs are the same as those found in serum/blood of lung fibrosis patients or not.

* Reviewer2 #15: Comments to the Author

*15. The conclusion is written well and nicely summarises the findings of this review. Figures and tables are nicely organised.

Answer: Many thanks for your careful review, positive comments, and affirming our research.

Reviewer 3’s Comments

* Reviewer3 #1: Comments to the Author

*1. Please use professional editing service to improve the review.

Answer: Many thanks for your careful review and positive comments. This manuscript was sent to a native English speaker again for modifying the language according to your comments. If you think the language level of the article is not good enough this time, you can inform us, and we will invite a language editing company to help us revise it again.

* Reviewer3 #2: Comments to the Author

*2. Very long sentences, mixed use of tense, incomplete sentences, and use of unusual words like non-negligible, makes the review very difficult to follow.

Answer: Many thanks for your careful review and positive comments. This manuscript was sent to a native English speaker again for modifying the language according to your comments. If you think the language level of the article is not good enough this time, you can inform us, and we will invite a language editing company to help us revise it again.

* Reviewer3 #3: Comments to the Author

*3. The titles for sections 4 and 5 make the reader feel that EV's are pathogenic in lung fibrosis. Change the heading, replace pathogenesis with therapy.

Answer: Many thanks for your careful review and positive comments.

The section 4 has been changed into “Contributions of EVs derived from mesenchymal stem cells (MSCs) in the treatment of pulmonary fibrosis” (Line 116-117)

The section 5 has been changed into “Contributions of EVs derived from macrophages in the treatment of pulmonary fibrosis”. (Line 203-204)

* Reviewer3 #4: Comments to the Author

*4. Bleomycin-induced pulmonary fibrosis model is cited several times. A brief paragraph on this model will be helpful.

Answer: Many thanks for your careful review and positive comments.

Bleomycin is a drug used to treat different types of neoplasms. Bleomycin most severe adverse effect is lung toxicity, which induces remodelling of lung architecture and loss of pulmonary function, rapidly leading to death.

A brief description of Bleomycin-induced pulmonary fibrosis model is added as your suggestions. (Line 170-173)

Reviewer 2 Report

This manuscript reviews the current knowledge in the field of EVs for pulmonary fibrosis and specifically the specific miRNA in EVs that are known to contribute to/ alleviate pulmonary fibrosis. To improve the manuscript I have included specific comments below.

Abstract

Line 20 – Not sure if ‘prototype’ is the best word to use here

Introduction - Gives a very general overview of pulmonary fibrosis (PF), more is required on the current gaps in knowledge and how this review fulfils the needs in the field.

Line 47 – should modify the sentence ‘Cell therapy is emerging…’ so that it’s more specific for PF.

Line 63 – should mention that EVs are classed not only size but also other factors, e.g. their biogenesis, specific markers, function.

Line 74 – ‘EVs as therapeutic targets for PF’? This is not a correct description as the manuscript is more on EVs as therapeutic ‘approaches’ for PF.

Line 64, 96 – Apoptotic bodies can be much smaller than 800nm – this should be corrected

Heading for Section 4 – ‘Contributions of EVs derived from MSCs in the pathogenesis of pulmonary fibrosis’ – should ‘pathogenesis’ be replaced with ‘treatment’?

Line 127 – the sentence starting with ‘Upon an acute lung injury condition..’ should be modified to ensure that correct message is conveyed – there are many causes of PF, often unknown, but acute lung injury (should of course specify the specific types of acute lung injury) can contribute to the development of PF. 

Line 173-184 – this section titled ‘Other MSC-derived EVs’ contain lots of useful information regarding how EVs can alter the course of PF. This is the section that differentiates this manuscript from a large volume of other papers that superficially mention how EVs can be used for the treatment of PF. Please could you expand more on how these EVs and the specific miRNAs named in this paper are used in what aspect of PF pathophysiology. Include what the authors’ views are in using these specific types of EVs. Then this review will be able to help future researchers working on new approaches to target PF and fulfil the purpose of this review ‘particular emphasis on the effects of EV-associated miRNAs’ as stated in line 74-75.

Section 5 is a critical part of this manuscript as it adds to the field much more than MSC-derived EVs (since there is a huge number of paper and knowledge on this). This section should be expanded further to include more details on the profibrogenic effects of EVs – e.g. what are the specific roles of the miR-328 inside EVs? Are there any other studies that report the role of the antifibrotic miR-142-3p in the field of PF?

Line 205 – include a brief description of Schisandra when it’s first introduced.

Line 213 – ‘extracellular vehicles’ should be corrected to ‘vesicles’

The title of Section 6 should be modified to better reflect the text/content in that section

Section 7 is the highlight of this manuscript, containing key information that differentiate this paper from other published papers. Consider moving Section 7 up towards the beginning of the manuscript, as it was quite difficult to wait until the end of the manuscript to find the content I anticipated from the title of the manuscript.

Line 344- when these EV-miRNAs play important roles in PF, are they eventually contributing to the pro-fibrotic processes or anti-fibrotic processes? Are these miRNAs the same as those found in serum/blood of PF patients?

The conclusion is written well and nicely summarises the findings of this review. Figures and tables are nicely organised.

Author Response

(The authors gave the same response as above.)

Reviewer 3 Report

Please use professional editing service to improve the review.

Very long sentences, mixed use of tense, incomplete sentences and use of unusual words like non-negligible, makes the review very difficult to follow.

The titles for sections 4 and 5 makes the reader feel that EV's are pathogenic in lung fibrosis. Change the heading, replace pathogenesis with therapy.

Bleomycin-induced pulmonay fibrosis model is cited several times. A brief paragraph on this model will be helpful.

Author Response

(The authors gave the same response as above.)

Round 2

Reviewer 1 Report

The authors answered most of my questions and I don't have further comments. 

Author Response

Xiuru Zhang, PhD, MD, MSc

Professor of Pathology, Department of Pathology

The Third People's Hospital of Longgang District

278 Songbai Road, Henggang Street, Shenzhen, China

Email: zhangxiurudr@126.com; Phone: +86–15350402147

Eden Wang, Ph.D.

Assistant Editor, Pharmaceuticals

Pharmaceuticals-1918912 “The emerging role of extracellular vesicles from mesenchymal stem cells and macrophages in pulmonary fibrosis: Insights into miRNAs delivery”

Dear Dr. Wang,

Thank you very much for sending us the reviews of our manuscript. The comments were all valuable and were very helpful for revising and improving our paper. According to the reviewers' and editor's comments and suggestions, we have revised the manuscript and responded, point by point, to the comments, as listed below. Revised portions are marked in light grey on the paper. All authors have read and approved the final version of the manuscript. We would like to thank you for considering our work for publication in Pharmaceuticals.

Yours sincerely,

Prof. Xiuru Zhang, PhD, MD, MSc.

(On behalf of all co-authors)

Replies to Reviewers and Editor

First, we thank both reviewers and the editor again for their careful review and valuable comments. The comments are all helpful for revising and improving our manuscript.

Reviewer 1’s Comments

* Reviewer1 #1: Comments to the Author

*1. The authors answered most of my questions and I don't have further comments. Answer: Many thanks for your careful review, positive comments, and affirming our research.

Reviewer 2 Report

The additional text added to the manuscript and the edited parts have surely improved the manuscript.  However, there are some points from the Reviewers that are not fully addressed by the authors:

-       In response to Reviewer 1 comment #8, the authors’ reference to the new text added in Line 48-50, Line 59-64, Line 66-72, Line 73-76 do not cover the advantages of EVs compared with cell therapy on the antifibrotic effects of EVs

-       The new sentence added in Line 70-72, mentioning the function of EVs decided by the composition of cargo instead of the size, might not be suitable to be added here. The function of EVs may be dependent on many other factors such as the microenvironment or external cues, so probably better to omit this sentence here.

-       The revised sentence in Line 137-138 seems out of place. Please make sure that it is within the right context.

-       Authors claimed that the correction from ‘extracellular vehicles’ to ‘extracellular vesicles’ was done but it has not been done (line 232)

Careful language editing is still required in some parts, e.g. 'Mechanically' is used a few times - do you mean 'mechanistically'?

Author Response

Xiuru Zhang, PhD, MD, MSc

Professor of Pathology, Department of Pathology

The Third People's Hospital of Longgang District

278 Songbai Road, Henggang Street, Shenzhen, China

Email: zhangxiurudr@126.com; Phone: +86–15350402147

Eden Wang, Ph.D.

Assistant Editor, Pharmaceuticals

Pharmaceuticals-1918912 “The emerging role of extracellular vesicles from mesenchymal stem cells and macrophages in pulmonary fibrosis: Insights into miRNAs delivery”

Dear Dr. Wang,

Thank you very much for sending us the reviews of our manuscript. The comments were all valuable and were very helpful for revising and improving our paper. According to the reviewers' and editor's comments and suggestions, we have revised the manuscript and responded, point by point, to the comments, as listed below. Revised portions are marked in light grey on the paper. All authors have read and approved the final version of the manuscript. We would like to thank you for considering our work for publication in Pharmaceuticals.

Yours sincerely,

Prof. Xiuru Zhang, PhD, MD, MSc.

(On behalf of all co-authors)

Replies to Reviewers and Editor

First, we thank both reviewers and the editor again for their careful review and valuable comments. The comments are all helpful for revising and improving our manuscript.

Reviewer 1’s Comments

* Reviewer1 #1: Comments to the Author

*1. The authors answered most of my questions and I don't have further comments. Answer: Many thanks for your careful review, positive comments, and affirming our research.

Reviewer 2’s Comments

* Reviewer2 #1: Comments to the Author

*1. In response to Reviewer 1 comment #8, the authors’ reference to the new text added in Line 48-50, Line 59-64, Line 66-72, Line 73-76 do not cover the advantages of EVs compared with cell therapy on the antifibrotic effects of EVs

Answer: Many thanks for your careful review and positive comments. We have added it now. Thanks again (Line 64-68)

* Reviewer2 #2: Comments to the Author

*2. The new sentence added in Line 70-72, mentioning the function of EVs decided by the composition of cargo instead of the size, might not be suitable to be added here. The function of EVs may be dependent on many other factors such as the microenvironment or external cues, so probably better to omit this sentence here.

Answer: Many thanks for your careful review and positive comments. I can not agree with you more. Thanks again. We have deleted this sentence. (Line 73-74)

* Reviewer2 #3: Comments to the Author

*3. The revised sentence in Line 137-138 seems out of place. Please make sure that it is within the right context.

Answer: Many thanks for your careful review and positive comments. We have deleted this revised sentence in original Line 137-138 as your suggestion. (Line 138)

* Reviewer2 #4: Comments to the Author

*4. Authors claimed that the correction from ‘extracellular vehicles’ to ‘extracellular vesicles’ was done but it has not been done (line 232).

Answer: Many thanks for your careful review and positive comments. We have revised it based on your suggestion. Meanwhile, we have checked all of them, and we confirmed that they have been revised to “extracellular vesicles”. Thanks again. (Line 231)

* Reviewer2 #5: Comments to the Author

*5. Careful language editing is still required in some parts, e.g. 'Mechanically' is used a few times - do you mean 'mechanistically'?

Answer: Many thanks for your careful review and positive comments. We have revised it based on your suggestion. Meanwhile, we have checked all of them, and we confirmed that they have been revised into “mechanistically”. Thanks again.

In addition, we bought a software (Grammarly) member to modify the language and grammar of the full text of the manuscript.

Reviewer 3’s Comments

* Reviewer3 #1: Comments to the Author

*1. The review is much improved. However, needs professional editing. Still has very long sentences. Thanks.

Answer: Many thanks for your careful review, positive comments, and affirming our research. This time we bought a software (Grammarly) member to modify the language and grammar of the full text of the manuscript. Meanwhile, could you please give us some specific suggestions, we will modify them one by one. Thank you very much.

Reviewer 3 Report

The review is much improved. However, needs professional editing. Still has very long sentences. Thanks.

Author Response

(The authors gave the same response as above.)
